# On the Tribocorrosion Behavior of Fe-Mn-Al-C Alloys in Ringer's Solution

Gisselle M. Barona-Osorio [1], Leonel A. Teran [2], Sara A. Rodríguez [1] and John J. Coronado [1,*]

1   Research Group of Fatigue and Surfaces, Mechanical Engineering School, Universidad del Valle, Cali 760032, Colombia
2   Department of Mechanical Engineering, Fundación Universidad de América, Bogotá 111711, Colombia
*   Correspondence: john.coronado@correounivalle.edu.co

**Abstract:** The long-term performance of steels is affected by the simultaneous actions of wear and corrosion, known as tribocorrosion. The tribocorrosion behavior of fully austenitic steels: Fe-Mn-xAl-C (x = 0, 3.5 and 8.3 wt.%) in Ringer's solution was investigated by using a pin on disk tribometer adapted with a three-electrode corrosion cell. Open circuit potential and coefficient of friction evolution as well as polarization curves were measured. Corrosion rates were calculated by the Tafel extrapolation method, and wear rates were calculated by using a linear profilometer. Pure and total wear rates were higher for the 3Al alloy due to the greater precipitation and embedded calcium minerals, hydroxides, and oxides on the surface, to the detachment of the deformed layer and its adhesion to the counterbody. Additionally, the 8Al alloy exhibited the lowest tendency to corrosion and corrosion rate and the greatest synergistic effect, indicating that this alloy is more sensitive to this effect than the other alloys. For the three materials, the change in the wear rate due to corrosion had a greater contribution to the synergy than the change in corrosion rate due to wear and the damage in the materials was derived mainly from pure mechanical wear.

**Keywords:** Fe-Mn-Al-C steels; tribocorrosion; Ringer's solution





## 1. Introduction

During the last two decades, Fe-Mn-Al-C alloys have been widely investigated for different industrial applications due to their low density and excellent combination of strength and ductility compared with conventional high manganese steels [1–5]. The outstanding combination of mechanical properties exhibited by these materials is associated with the deformation mechanisms, which are correlated with the austenite stabilization and the stacking fault energy (SFE) [6–10]. The latter concept is defined as the energy cost for a transformation to occur by means of plastic deformation of the material and depends mainly on composition and deformation temperature [11]. Alloying elements such as manganese and aluminum modify the SFE values of this family of alloys [12,13]. In the case of aluminum, the addition of this element increases the SFE value and, in turn, suppresses or delays the appearance of mechanical twins as reported by Park et al. [14], which was reflected through variations in yield and tensile strength.

SFE value and the strain hardening behavior are related to the wear resistance of Fe-Mn-Al-C alloys. Abassi et al. [15] compared the abrasive wear behavior of two Hadfield steels with and without aluminum. They found that steel without Al showed better wear resistance under high stress conditions due to twinning deformation and dynamic strain aging, which lead to microstructural changing during the wear process. In another study, Zambrano et al. [8] also compared the abrasive wear behavior under different applied loads of three austenitic steels: an Fe-Mn-Al-C, a Hadfield and an AISI 316L in terms of their SFE values. They found that the wear resistance coefficient, k, increased with the increase in the load for the three alloys and noticed a transition in wear mechanism from microploughing

to microcutting as load increased. They also observed that the steel with the lowest SFE showed the highest wear resistance due to the rapid strain hardening behavior and the formation of martensite laths, which retards the plastic deformation by the accumulation of dislocations in these regions.

On the other hand, the action of the synergism between wear and corrosion, known as tribocorrosion, is responsible for damages to machines and devices in different areas of engineering [16,17]. In particular, implants are continuously exposed to wear caused by friction in joints in the presence of corrosive solutions (body fluids) which affect their long-term performance [18]. Additionally, the knowledge of the tribological behavior in the absence of a corrosive medium and of the electrochemical behavior in the absence of wear is not sufficient to predict the behavior of their synergism [19,20]. Thus, the study of the synergy between wear and corrosion is of great importance to understand and improve the tribocorrosion behavior response of metallic materials, mainly in the areas of orthopedic surgery and dental surgery because it has a high impact on health and economics [21].

Zhang et al. [22] studied the electrochemical corrosion behavior of Fe-Mn-Al alloys with variations in Al content between 0–5 wt.% in different aqueous solutions. They found that by adding Al, an increase in corrosion potential ($E_{corr}$) and a decrease in corrosion current density ($i_{corr}$), were obtained. However, after the polarization test in the saline solution, pitting was observed on the surface of the 5% Al sample and no surface passivation was observed. The behavior was attributed to the capacity of the Al to the formation of a stable oxide due to its high passivation coefficient and standard electrode potential ($-1.66$ V vs. SHE, standard hydrogen electrode), which is more negative than that of the Mn content ($Mn^{2+}$: $-1.18$ V vs. SHE; $Mn^{3+}$: $-0.283$ V vs. SHE).

The corrosion properties of two Fe-25Mn-(5,8)Al-0.2C and an Fe-30Mn-4Al-4Cr-0.2C dual phase steels in 30% $HNO_3$ and 3.5 wt.% solutions were investigated by Hamada et al. [23]. They found that the alloy with Cr additions showed a stable passive region at a high positive potential and the lowest passivation current density in 30% $HNO_3$ solution due to the presence of Al and Cr, which induce the formation of a high stability passive film by direct nucleation and growth mechanism on the steel surface, where an enrichment of Al and Cr and a depletion of Mn and Fe atoms occurred. Regarding the saline solution, they found that the three steels showed a similar polarization behavior, and the main cathodic reaction was oxygen reduction. When the potential increased towards the anodic regime, the dissolution of the alloying elements of Mn and Al as well as matrix Fe would occur. Finally, they improved the corrosion resistance by aging through an anodic passivation treatment in $HNO_3$ acid, due to the Al nucleation and growth mechanism.

Chen et al. [24] and Hsueh et al. [25] investigated the properties of Fe-Mn-Al-C alloys after nitriding treatments for several hours. They observed the formation of a layer of aluminum nitrides (AlN) with fcc structure on the surface of the material. The layer was responsible for the observation of a passivation zone and a decrease in current density, which results in an improvement in the corrosion resistance of the material and the inhibition of pitting corrosion. Regarding the mechanical properties, they concluded that these nitrides increase the hardness of the surface, the yield stress and total elongation when comparing with untreated alloy. Contrasting with the literature, they concluded that nitrited Fe-Mn-Al-C alloys are potential candidates to replace AISI 316L for biomedical purposes due to their excellent mechanical reliability and electrochemical stability. Moreover, Villamizar et al. [26] studied the corrosion and wear behavior of Fe-Mn-Al-C alloys as substrate with multilayer coating deposition of Ti alloys. Those multilayer systems were subjected to the simultaneous action of wear and corrosion in Hank's solution. However, in their work they only show polarization curves and wear tracks of the two multilayer systems, but they did not show synergistic results of wear and corrosion or the comparison of the results of multilayers with the substrates (Fe-Mn-Al-C matrix).

To the author's knowledge, few investigations have been reported concerning the wear and corrosion behavior of Fe-Mn-Al-C alloys [1,5,22,27]. In addition, currently there are research projects involving both phenomena acting independently from each other, mainly

with heat treatments or coatings on the surface to improve their response [24,25]. Therefore, the aim of this study was to investigate the tribocorrosion behavior of three solution-treated austenitic Fe-Mn-xAl-C (x = 0, 3.5 and 8.3 in wt.%) alloys in Ringer's solution to elucidate the role of Al additions in the wear–corrosion synergism. For this purpose, a device was designed and adapted to evaluate and quantify the tribocorrosion phenomenon. In the present system, the total wear rates increased with respect to the pure wear rates indicating that corrosion reduces the wear resistance. For all materials the main contribution to the total wear rate was due to pure sliding wear. Moreover, the largest contribution to the synergism corresponded to the change in the wear rate due to corrosion. Finally, in this case of sliding wear in combination with corrosion and the addition of Al (which modifies the SFE), the formation of precipitates and oxides influences the total wear behavior.

## 2. Materials and Methods

Ingots of three Fe-Mn-xAl-C (x = 0, 3.5 and 8.3 in wt.% and hereafter called 0Al, 3Al and 8Al alloys) were obtained by induction melting by means of high purity Fe, Mn, Al and Fe-4C. After melting process, ingots were hot rolled at 1200 °C by applying reduction steps of 2% (80 steps) to plates of 6 mm of thickness and then were cooled in air. Subsequently, samples were solution-treated at 900 °C for 1 h followed by furnace cooling to obtain a fully austenitic structure with isotropic properties and to relieve the stresses produced during the plastic deformation process of the material. The chemical composition of the alloys was carried out by means of spectrochemical analysis and is shown in Table 1. The density was measured by means of high precision hydrostatic balance ($\pm$0.0001 g/cm$^3$) based on Archimedes' principle.

**Table 1.** Chemical composition of the alloys.

| Alloy | Composition (wt.%) | | | |
|---|---|---|---|---|
| | **Mn** | **Al** | **C** | **Fe** |
| 0Al | 20.5 | 0.0 | 0.87 | Balance |
| 3Al | 22.2 | 3.5 | 0.84 | Balance |
| 8Al | 22.1 | 8.3 | 0.89 | Balance |

Flat square samples of 25 × 25 × 5 mm$^3$ were obtained in the rolling direction by machining for the tribocorrosion tests. The microstructures of the samples were characterized by using an Optical Microscope (OM) and X-ray Diffraction (XRD). The Vickers microhardness tests were performed with a Vickers Tester TH717 wherein a 0.98 N indentation load and time of 15 s of holding load were used.

### 2.1. Tribocorrosion Test

The experiments were carried out by adapting a pin on disk tribometer with a conventional three-electrode cell where the flat square sample was used as the working electrode, an Ag/AgCl electrode was used as reference electrode and stainless steel was used as the counter electrode. Commercial ZrO$_3$ spheres of 6 mm of diameter were used as counterbody (E = 175 GPa, ν = 0.27 provided by the supplier).

Ringer's solution reported in other works [28,29] with a pH of 7.8 $\pm$ 0.2 was used as the electrolyte (Table 2 shows the composition of the solution).

**Table 2.** Composition of the Ringer's solution (g/L).

| Composition | Weight (g) |
|---|---|
| NaCl | 9.00 |
| KCl | 0.43 |
| CaCl$_2$ dihydrate | 0.24 |
| NaHCO$_3$ | 0.20 |

Before the tests, all samples were taken to a mirror-like finish starting with #400 sandpaper up to #1200 and then cloth polished using 0.3 μm alumina suspension to obtain homogeneous surfaces. The surface roughness was obtained across to the rolling direction with a Mitutuyo SJ–201P roughness tester in which a cutoff length of 0.25 mm, a sampling length number of 5 and displacement speed of 0.25 mm/s were used. Each test was performed three times. The difference in the mean roughness for all the samples was not significantly different and its value was $R_a = 0.029 \pm 0.007$ μm. The samples were washed in ultrasonic cleaner with alcohol during ten minutes prior to immersion; the exposed surface area was a ring of about 1.90 mm$^2$ (internal radius of 6.6 mm and external radius of 10.2 mm) while the other parts of the samples were coated with a layer of paraffin wax.

The test rig consisted of the following parts (Figure 1 shows the adapted device):

1.  Level arm with a stem to hold and position the counterbody to the desired radius.
2.  Counterweight to balance the level arm in a horizontal position without any load.
3.  Disc, which was attached to a rotating shaft.
4.  Corrosion cell was coupled to the rotating disc by means of clamping screws. At the bottom of the cell, the working electrode (test sample) was held by means of fastening clips and screws that were located on four lines at 90° of threaded holes on the cell bottom surface (Figure 2a); the whole part (test sample plus corrosion cell) presented the rotational movement. To enhance the adjustment of the cell to the shaft and avoid subsequent wear of the cell, threaded bronze sleeves were adapted on the lower face of the cell.
5.  Load cell holder to hold the load cells and to measure the tangential force generated by the tribological pair.
6.  Counterbody holder to hold the ZrO$_3$ ball.
7.  The last part of the device corresponds to the electrical adaptation in which another device was built to keep the electrical contact between the potentiostat and the working electrode during the tests (Figure 2b). The main element to transmit the electric current was an axial bearing. A wire was welded to the bearing in the moving component to make contact to the working electrode and another one to the static component to link to the potentiostat. To improve the electrical conductivity between both parts, conductive graphite grease was applied periodically.

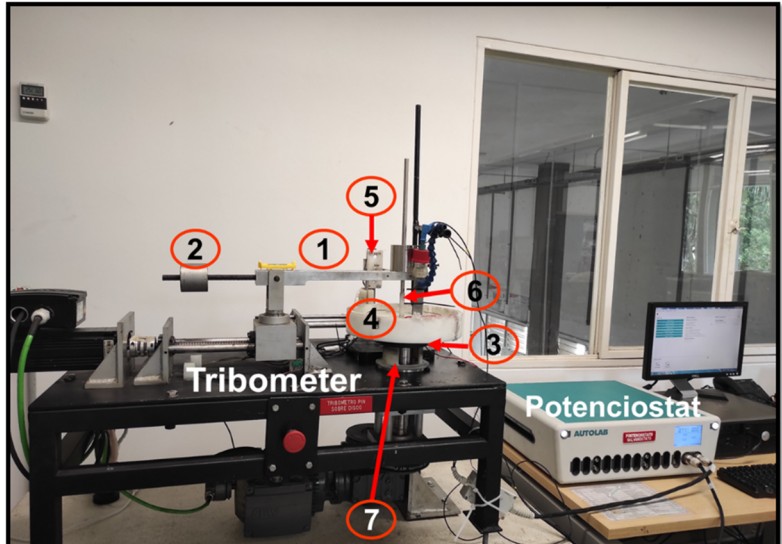

**Figure 1.** Final setup of the device. 1. Level arm; 2. Counterweight; 3. Disc; 4. Corrosion cell; 5. Load cell holder; 6. Counterbody holder and 7. Electrical adaptation.

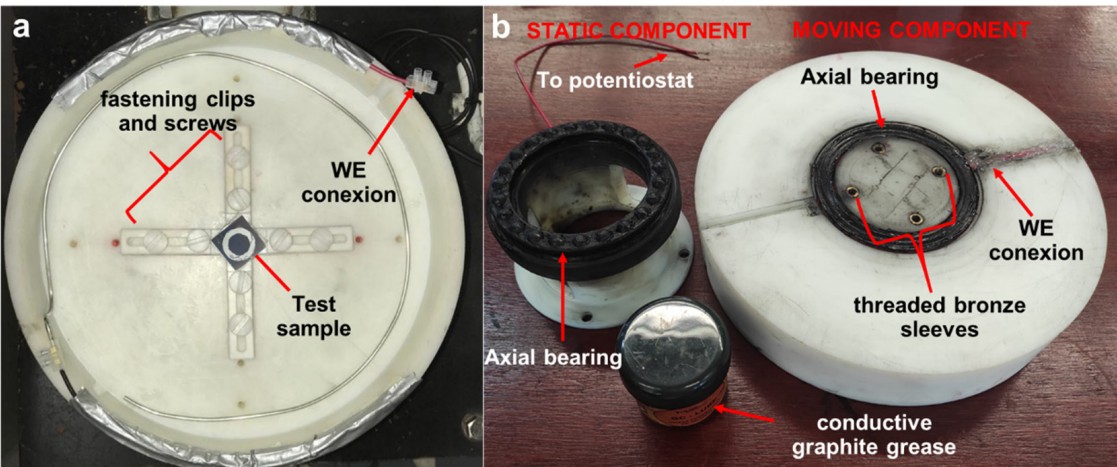

**Figure 2.** (**a**) Front; (**b**) back of corrosion cell.

The storage capacity of the cell is 1.3 L. The corrosion cell, sample and counterbody holders were made with Nylon to ensure electrical insulation.

According to the ASTMG119 [30], the synergism, S, between wear and corrosion is described in terms of metal loss due to both mechanical and chemical processes by means of the following Equation (1):

$$S = T - W_0 - C_0, \tag{1}$$

where T is the total degradation rate of the material due to wear and corrosion acting together, $W_0$ is the wear rate without the action of corrosion and $C_0$ is the corrosion rate when there is no wear. Since wear affects corrosion and vice versa, total synergy can be written as the sum of two components, the change in the rate of mechanical wear due to corrosion, $\Delta W_c$, and the change in the corrosion rate due to mechanical wear, $\Delta C_w$ as follows (2):

$$S = \Delta C_w + \Delta W_c, \tag{2}$$

where $\Delta C_w = C_w - C_0$ and $\Delta W_c = T - (W_0 + C_w)$, here $C_w$ is the corrosion rate with the action of wear. Finally, with the steps described below, T, $W_0$, $C_0$ and $C_w$ are measured to calculate S (based on [30]).

1. *Corrosion test to measure the corrosion rate*, $C_0$: the samples were immersed into the solution during 20 min until a stable open circuit potential (OCP) was obtained. After this time, a potential sweep of 200 mV was applied above and below the OCP value (OCP $\pm$ 200 mV) at step size of 0.15 mV and at a scan rate of 0.16 mV/s according to the ASTM G5 standard [31]. From this measurement, corrosion current density ($I_{corr}$) and corrosion potential ($E_{corr}$) were obtained by means of the Tafel extrapolation method. The values were computed by means of linear fitting. The rules that were applied to the extrapolation methods according to some authors [32,33]: at least one of the branches of the polarization curve should exhibit Tafel behavior over at least one decade of current density and the extrapolation should start at least 50 to 100 mV away from $E_{corr}$.

2. *Pure sliding wear test under cathodic protection to obtain the wear rate*, $W_0$: a load of 10 N was applied to the sphere–sample contact, which corresponds to mean pressures of 1304, 1287 and 1271 MPa estimated using the Hertz contact theory and corresponding to an initial contact radius about 0.0605, 0.0608 mm and 0.00613 mm for the 0Al, 3Al and 8Al, respectively. However, these contact pressures are enough to produce plastic deformation of the material. Tangential speed was 2 mm/s and the positioning radius of the counterbody was fixed in about 8 mm.

To perform this test, the working electrode initially was held at a potential of −0.5 V with respect to $E_{corr}$ and current behavior was measured during 20 min. Subsequently,

wear test was started and was carried out during 1 h. When the wear test finished, current behavior was measured during the same time interval as the beginning.

To calculate the wear rate of the materials, the cross-sectional area of the worn track was measured by profilometry using a KLA Tencor D-120 contact profilometer. The scanning speed was 10 mm/s making a matrix of 20 paths, each of length 1.5 mm, in steps of 0.1 mm, and with contact force of 10 mg.

3.  *Wear-corrosion test to find the total wear rate*, T: the wear test described in item (2) was repeated without cathodic protection, i.e., with the test sample exposed to the electrolyte without applying any potential and the evolution of the open circuit potential was studied without the polarization of the sample. The OCP recording was started and measured during 20 min before starting the wear test. At the end of the wear test, the behavior of the OCP was measured during the same time interval as the beginning.

4.  *Wear-corrosion test to find the corrosion rate under the action of wear*, $C_w$: the tests described in items (1) and (2) were repeated without cathodic protection. The wear test was started and the OCP was allowed to stabilize for 20 min (the time should be long enough to allow the potential to stabilize, but it should not be long either to avoid excessive wear of the sample). After 20 min, the linear polarization test was started and at the end, the sample was removed to avoid further loss of dimensions and for further analysis. From this measurement, corrosion current density ($I_{corr}$) and corrosion potential ($E_{corr}$) were obtained by means of Tafel extrapolation method.

All experiments were carried out at room temperature. The synergy between wear and corrosion was calculated by using Equation (1), and from Equation (2), $\Delta W_c$ and $\Delta C_w$ were calculated.

Scanning electron microscopy (SEM) was used to characterize the worn surfaces, and to obtain the hardening profiles, micro-hardness tests were performed on the cross section of the samples. For these tests, a matrix of $10 \times 5$ indentations was made starting with a distance of 20 μm from the worn area. The spacing between each indentation mark were three times the value of the diagonal measured for each sample.

### 2.2. Statistical Analysis

Each test was performed three times and the results were reported with a confidence interval calculated at 95%. For the case of corrosion rates, additional tests (four tests) were carried out. One-way ANOVA analysis with a significance level of 0.05 and a Tukey test were used for statistical analysis of the results.

## 3. Results and Discussion

### 3.1. Microstructural Characterization

Figure 3 shows the optical micrographs of the three Fe-Mn-xAl-C alloys after thermomechanical and solution treatments. For all materials, a characteristic microstructure of austenite with annealing twins was observed. Homogeneous equiaxial grains were observed for the 0Al alloy, while for the 3Al and 8Al alloys, some elongated grains and equiaxial grains were observed as a consequence of the thermomechanical treatment and the solution treatment, respectively. To obtain the grain size, the recommendations of the ASTM E112-12 [34] were followed. From a 100× picture of the microstructure of the material, a random line was drawn on the image, then a count of how many grains were intercepted by the line was carried out and finally the value was divided by the line length to obtain the average size, this procedure was performed four times in longitudinal and transversal directions. The grain sizes of the alloys were $63 \pm 21$, $52 \pm 2$ and $49 \pm 7$ μm for the 0Al, 3Al, and 8Al, respectively. It was determined that the differences between the grain sizes for the three Fe-Mn-xAl-C alloys were not significant; therefore, the grain size was kept constant between the alloys in order to eliminate the possible effect it may have on the tribological behavior. Some casting defects (black spots) were also found near the grain boundaries and in some areas within the grain for the 0Al and to a lesser degree for the

3Al and 8Al, which were identified by using Energy Dispersive X-ray Spectroscopy (EDS) as microporosities and iron or manganese oxides, as well as pitting caused by chemical attack. Figure 4 and Table 3 show the results, here, the displayed numbers correspond to the spectrum.

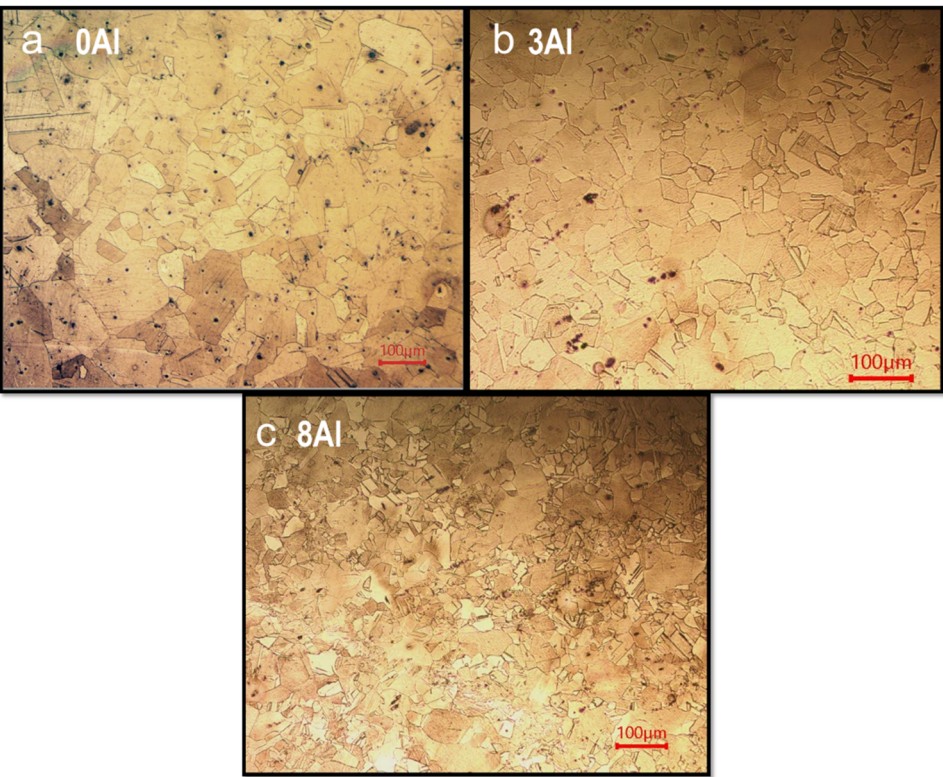

**Figure 3.** Microstructures at 100× of the (**a**) 0Al; (**b**) 3Al; (**c**) 8Al alloys.

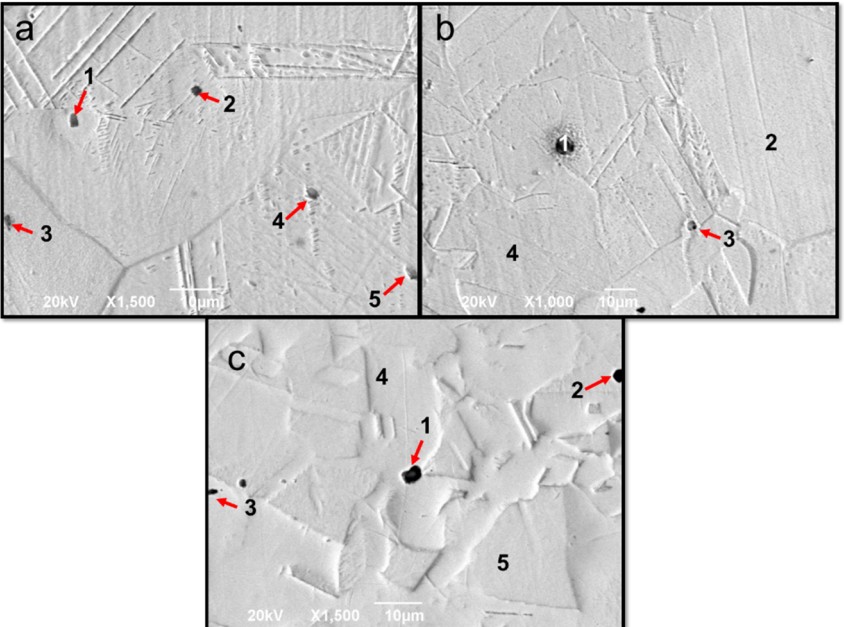

**Figure 4.** SEM images of the microstructures of the (**a**) 0Al; (**b**) 3Al; (**c**) 8Al alloys. The numbers correspond to the spectrum showed in Table 3.

**Table 3.** EDS composition (wt.%) corresponding to the surfaces of the alloys shown in Figure 4.

| Alloy | Spectrum | O | S | Mn | Al | Fe | Mo | N |
|---|---|---|---|---|---|---|---|---|
| | 1 | 11.30 | - | 18.18 | - | 68.61 | 1.91 | |
| | 2 | 10.57 | 0.69 | 19.73 | - | 69.01 | - | |
| 0Al | | 12.01 | 0.83 | 19.18 | | 67.99 | - | |
| | | 8.89 | - | 18.73 | | 72.37 | - | |
| | 3 | 8.20 | - | 18.79 | | 73.01 | - | |
| | 1 | 11.95 | - | 13.58 | 27.91 | 39.96 | - | 6.60 |
| 3Al | 2 | - | - | 22.77 | 2.56 | 74.67 | - | - |
| | 3 | 11.38 | - | 19.19 | 4.27 | 65.16 | - | - |
| | 4 | - | - | 24.87 | 2.94 | 72.19 | - | - |
| | 1 | 10.21 | - | 12.56 | 28.80 | 38.41 | - | 10.02 |
| | 2 | 5.80 | - | 20.73 | 10.25 | 63.22 | - | - |
| 8Al | 3 | - | - | 22.13 | 14.10 | 63.77 | - | - |
| | 4 | - | - | 23.73 | 7.64 | 68.63 | - | - |
| | 5 | - | - | 22.30 | 8.37 | 69.33 | - | - |

Concerning the density, the results were $7.83 \pm 0.02$, $7.43 \pm 0.06$, and $7.03 \pm 0.06$ g/cm$^3$ for the 0Al, 3Al, and 8Al, respectively. Taking the 0Al alloy as a reference, reductions of 5% and 10% were observed for the 3Al and 8Al alloys, respectively. Frommeyer et al. [35] found density reductions in Fe-Al alloys greater than 10% when varying the Al composition between 0 and 8.5%. The density reduction with the increase in aluminum content is due to Al having a lower density and atomic weight than Fe (density of 2.7 g/cm$^3$ compared to 7.8 g/cm$^3$, respectively).

### 3.2. Hardness and Micro-Hardness

Table 4 shows the macro and micro-hardness of the materials, capital letters indicate the group according to ANOVA analysis. As the aluminum content increases, a slight increase in hardness was observed; however, for the 0Al and 3Al alloys, the values were not significantly different.

**Table 4.** Macro and micro-hardness of the materials.

| Alloy | Hardness (HV$_{196.1 \text{ N/15 s}}$) | Micro-Hardness (HV$_{0.98 \text{ N/15 s}}$) | Group [1] |
|---|---|---|---|
| 0Al | $201 \pm 7$ | $325 \pm 13$ | A |
| 3Al | $207 \pm 2$ | $333 \pm 12$ | A |
| 8Al | $280 \pm 9$ | $378 \pm 11$ | B |

[1] Groups according to ANOVA analysis.

Increases in hardness values with aluminum content have been reported by Sutou et al. [36] for Fe-20Mn-(10-14)Al-(0-1.8)C and Fe-20Mn-(10-14)Al-(0.75-1.8)C-5Cr alloys (wt.%). They ascribe this behavior to the formation of kappa carbides during the cooling of the materials in air and to the high content of aluminum and carbon. In the case of the study alloys, hardening by kappa carbides would not be possible since the Al content is less than 8.5%, and also the C content is less than 1% [27,37,38]. This behavior can relate to a solid solution hardening during the substitutional diffusion of Al atoms during the solubilization process as reported by Choi et al. [39]. Based on the literature, the authors obtained an expression for the minimum critical shear stress for dislocation slip to occur (plastic deformation) depending on the composition. They reported a value of the aluminum effect by solid solution in Fe-Mn-Al-C TWIP steels of 20 MPa per wt.% of Al addition. The increase in hardness is also related to the peaks shift of the austenite due to the Al addition found by the authors in a previous research [40]. An increase in the lattice parameter and the crystallite size occurred since Al is a lighter atom and has a

larger atomic radius which produces a diffusion in the crystalline lattice and substitutional replacement of Fe and Mn atoms, hence producing a widening of the lattice. However, the aforementioned hypotheses need further investigations.

The stacking fault energy of the steels were estimated at room temperature by using the X-ray diffraction (XRD) technique, described in a previous study [40]. It was found that the three alloys showed diffraction peaks corresponding to the austenite phase. In addition, it was noticed that the higher the aluminum content, the higher the SFE value; the values were $17.53 \pm 2.47$, $35.61 \pm 4.76$ and $50.76 \pm 6.73$ for the 0Al, 3Al, and 8Al, respectively. Lehnhoff et al. [41] and Park et al. [14] also reported SFE values in Fe-Mn-Al-C-Ni-N-Cr-Si alloys (Al content up to 2.5%) and Fe-Mn-Al-C (Al content up to 6%), respectively, which increased with the increase in Al content. Although the present alloys were designed in terms of their composition to have SFE values that would provide different deformation mechanisms, i.e., Transformation Induced Plasticity (TRIP), Twinning induced plasticity (TWIP), and Microbands Induced Plasticity (MBIP), according to the literature [8,9,35,39,42,43] it is possible that the three alloys are within the same TWIP deformation mechanism or have combined mechanisms. This is because in a recently work it was identified that there is a substantial dependence between the estimated value of SFE and the elastic constants chosen [40].

### 3.3. Corrosion Measurements, $C_0$

Figure 5 presents the electrochemical polarization curves of the materials immersed in Ringer's solution and Table 5 shows the corrosion potential values (compared to the Ag/AgCl reference electrode) and corrosion rates, capital letters indicate the group according to ANOVA analysis. The results show that corrosion potential values are significantly different for all the materials and that addition of Al increased (towards less negative values) the corrosion potential, indicating that the 8Al alloy has lowest tendency for the corrosion process. It was also found that the 8Al alloy showed lower corrosion rate than the 0Al alloy.

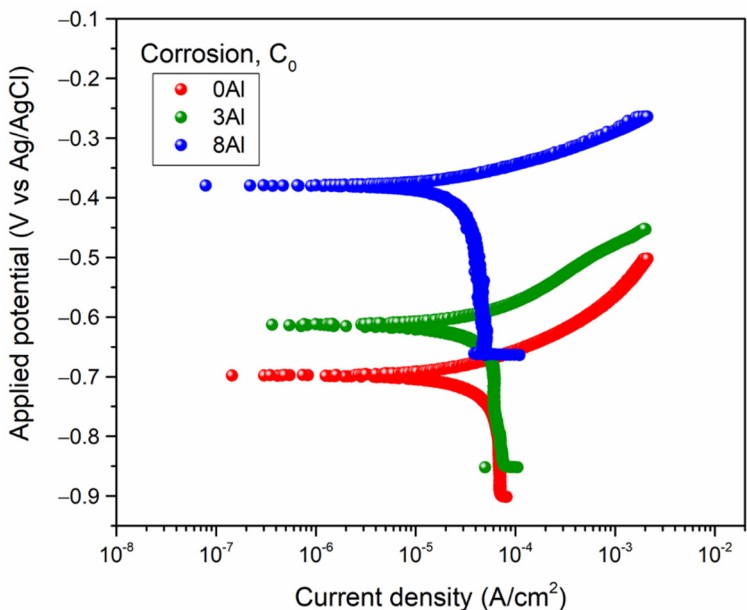

**Figure 5.** Polarization curves for the alloys corresponding to $C_0$ tests.

**Table 5.** Corrosion potential values (compared to the Ag/AgCl reference electrode) and corrosion rates of the materials.

| Alloy | Corrosion Potential $E_{corr}$ (V vs. Ag/AgCl) | Group [1] | Corrosion Rate ($mm^3/mm^2$ yr) | Group [1] |
|---|---|---|---|---|
| 0Al | $-0.70 \pm 0.02$ | A | $0.34 \pm 0.07$ | A |
| 3Al | $-0.59 \pm 0.04$ | B | $0.32 \pm 0.09$ | A, B |
| 8Al | $-0.38 \pm 0.07$ | C | $0.22 \pm 0.04$ | B |

[1] Groups according to ANOVA analysis.

From the polarization curves showed in Figure 5, it can be seen that during the cathodic reaction the limiting current density was achieved for all cases. Hence, the behavior of $E_{corr}$ observed for the materials is due to variations in the anodic reaction and is associated with changes in the equilibrium potential, the electrical double layer and the microstructure of the alloys. The Al additions modifies the equilibrium potential and the exchange current density of the anodic reaction owing to its more negative standard potential when comparing with Fe and Mn. Moreover, from Figure 3, it can be seen that the 0Al alloy exhibited a higher amount microstructural irregularities than the other two alloys, which is evidenced in the change in the equilibrium potential, since Al reduce the ability to form other type of oxides due to its increased negative Gibbs free energy. Similar results were found by Dieudonné et al. [44] for Fe-18Mn-0.6C alloys with the addition of 1.5% Al immersed in a 5% NaCl saline solution. In addition, Al has a high passivation coefficient, which indicates that it could form stable oxides on the surface [22], however, for this case, the Al content is not enough to produce such a stable layer over the whole surface and the corrosion process is governed by Fe and Mn dissolution.

*3.4. Pure Sliding Wear Measurements, $W_0$*

No significant differences in stabilization times (around 7 min) or steady state values of friction coefficient ($0.26 \pm 0.02$) were observed between the alloys during the pure sliding wear test. Figure 6 shows the wear rates of the materials under pure sliding wear and the micro-hardness on the wear tracks measured after the wear tests next to the symbol. For the 0Al and 8Al alloys, there were no significant differences between their values, and the 3Al alloy exhibited the highest wear rate. The micro-hardness on the wear tracks measured after the wear tests shows that the 3Al alloy presented the lowest micro-hardness, which is related to its highest wear rate.

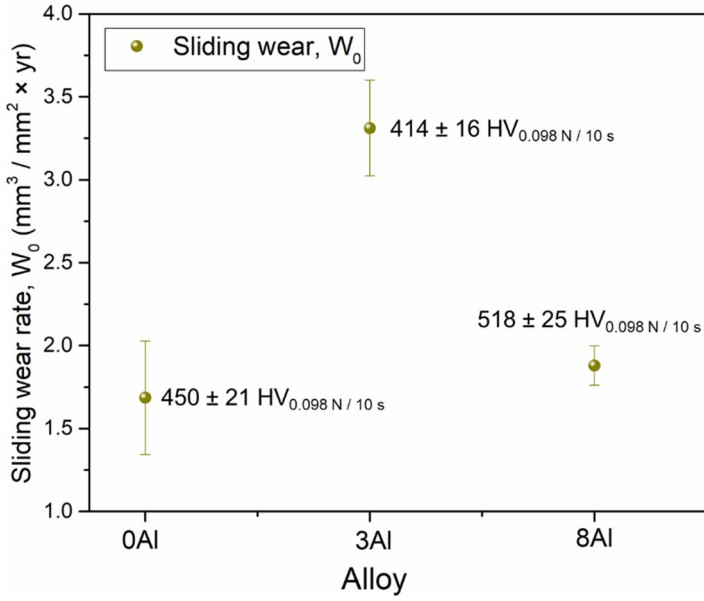

**Figure 6.** Wear rates of the materials under pure sliding wear, $W_0$.

From SEM images showed in Figure 7, the numbers correspond to the spectrum (see Table 6). It was found that particles were deposited on the surfaces of all the materials. By using EDS (Table 6), the presence of calcium and oxygen was observed, which are related to deposits of carbonates, oxides, and hydroxides; additionally, for the 3Al alloy, the presence of Cl was identified, indicating that additional deposits of the chloride mineral were formed. Zones with the presence of elements corresponding to the test bodies were observed on the counterbody surfaces. In the case of the 0Al and 3Al, iron content was found, and for the 8Al, iron, aluminum, and manganese were observed. The results indicate that there was adhesion of the test material to the counterbody. Additionally, the presence of Ca and C, some cracks, and detachment of the material loosed from hardened layers adhered to the counterbody were also observed.

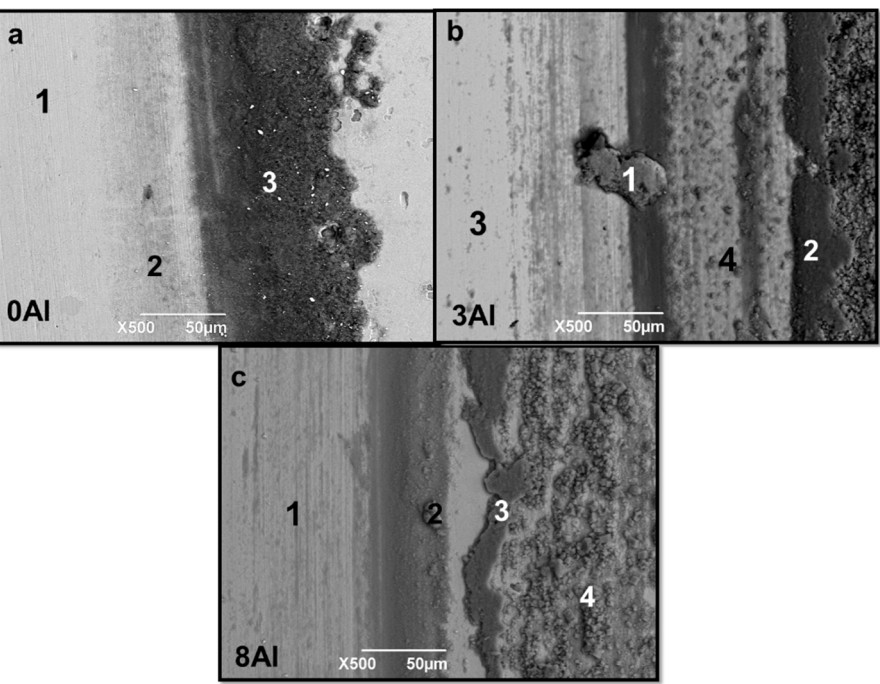

**Figure 7.** SEM images of the (**a**) 0Al; (**b**) 3Al; (**c**) 8Al alloys under pure sliding wear, $W_0$. The numbers correspond to the spectrum showed in Table 6.

**Table 6.** EDS composition (wt.%) corresponding to the surfaces of the alloys shown in Figure 7.

| Alloy | Spectrum | C | O | Al | Cl | Ca | Mn | Fe |
|---|---|---|---|---|---|---|---|---|
| 0Al | 1 | 7.88 | 6.94 | - | - | - | 20.97 | 64.30 |
|  | 2 | 18.07 | 10.37 | - | - | - | 20.28 | 63.26 |
|  | 3 | 23.23 | 36.39 | - | - | 19.80 | 5.62 | 14.96 |
| 3Al | 1 | - | 36.35 | - | 14.03 | 1.63 | 9.56 | 38.43 |
|  | 2 | - | 51.17 | - | 3.63 | 31.53 | 2.75 | 10.92 |
|  | 3 | 5.57 | - | 2.35 | - | - | 21.22 | 70.87 |
|  | 4 | 16.97 | 22.03 | 1.17 | 0.39 | 8.11 | 12.71 | 38.62 |
| 8Al | 1 | - | 8.34 | 5.60 | - | - | 19.12 | 66.94 |
|  | 2 | 12.89 | 54.11 | - | - | 33.00 | - | - |
|  | 3 | 11.98 | 58.56 | - | - | 29.56 | - | - |
|  | 4 | 14.83 | 36.01 | - | - | 20.66 | 7.36 | 20.14 |

The particles deposited on the surfaces of the materials after the pure sliding wear test are the result of the cathodic protection during the test, which produces a reduction in water and the production of $OH^-$ ions. In terms of the cathodic potentials, water reduction with the formation of hydroxide ions take place according with the following reaction:

$2H_2O + 2e^- \rightarrow H_2 + 2OH^-$. This phenomenon generates an increase in the local pH and modifies the inorganic balance of the electrolyte adjacent to the surface of the metallic material, which results in the formation of calcareous deposits and hydroxides on the metal surface. The hydroxide ions react with the $Ca^{2+}$ ions and a layer of $CaCO_3$ is formed on the surface in accordance with the applied potential [45]. This surface layer provides a porous physical barrier between the electrolyte and the steel surface [46]. Na and K ions are also present in the solution due to the composition of Ringer's solution, which could lead to the formation of sodium and potassium precipitates, however, from results shown in Table 6, only the presence of Ca and Cl were observed. The greater precipitation and embedding of particles on the surface in the 3Al alloy, as well as the debris detached from the material adhering to the counterbody producing possible abrasion and greater material detachment can explain its high wear rate.

### 3.5. Total Wear Measurements, T

Figure 8 shows the evolution of the coefficient of friction and open circuit potential (OCP) for the materials. In all the curves, a different behavior of both the coefficient of friction and the OCP was observed. For the 0Al, the OCP maintained a similar behavior before, during and after wear due to the fact that a protective layer is not formed but other types of unstable oxides are formed on the surface of the alloy. Additionally, in the initial period, an increase in OCP occurred due to the early contact between asperities and their breaking, which increases the contact area and modifies the anodic and cathodic zones on the surface, producing a more noble value. The irregular behavior (with fluctuations) of the coefficient of friction is attributed to the formation and expulsion of wear debris and oxides. For the 3Al alloy, initially a gradual drop in OCP was observed, followed by a more stable behavior as was also observed for the coefficient of friction.

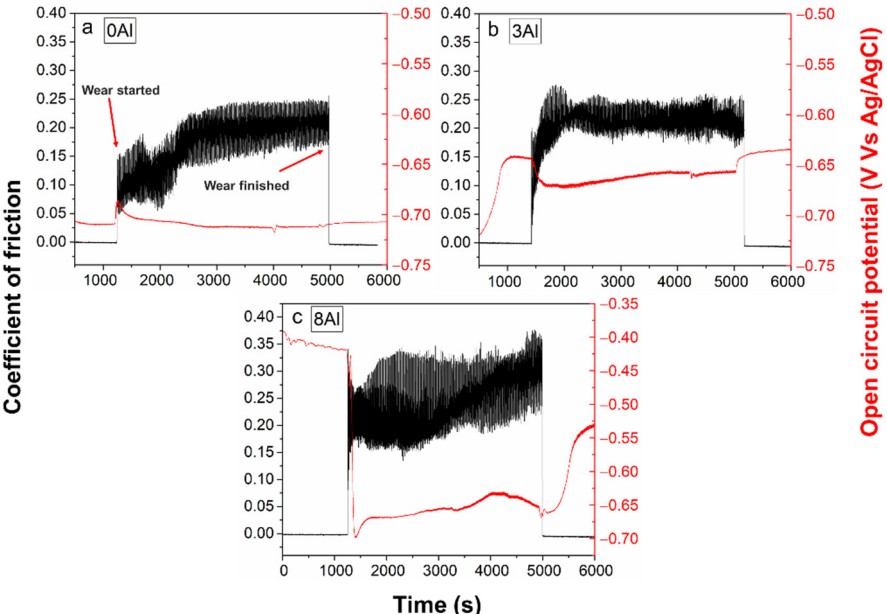

**Figure 8.** Evolution of the coefficient of friction and open circuit potential for the (**a**) 0Al; (**b**) 3Al; (**c**) 8Al alloys under total wear test, T.

For the 8Al alloy, the OCP dropped drastically when wear began and, additionally, other drops were observed at subsequent times, which is explained by the destruction of the passive layer and the initial contact of some asperities, which are deformed and produce an increase in the contact area thus generating some more-active zones. Subsequently, an increase in OCP was again observed up to around 4300 s during sliding. For this alloy, neither the OCP nor the coefficient of friction stabilized during the wear process, indicating a greater synergy effect on the alloy than the 0Al and 3Al alloys. Finally, the OCP tended

towards its initial value when the wear test was ended for the 3Al and 8Al alloys, indicating that there is a tendency to form a stable oxide layer on the surfaces of the material that protects them from corrosion.

The surfaces of the wear samples after the tribocorrosion test are shown in Figure 9, the numbers correspond to the spectrum (see Table 7). Particles with different morphologies were observed on the surfaces of the 0Al and 3Al alloys, which are mainly circular, are agglomerated in some areas, and can act as lubricant. By using EDS (Table 7), the presence of oxygen, manganese, iron and in some areas a lower proportion of chlorine and sodium was observed for the 0Al alloy, which corresponds to iron or manganese oxides and sodium chloride precipitates from the electrolyte. Moreover, for the 3Al and 8Al, the presence of aluminum was observed is some areas, indicating possible formation of aluminum oxide. The 3Al alloy showed the highest oxides proportion. Similar to $W_0$ study, material transfer to the counterbody was observed, indicating that the main wear mechanism was adhesion along with corrosion.

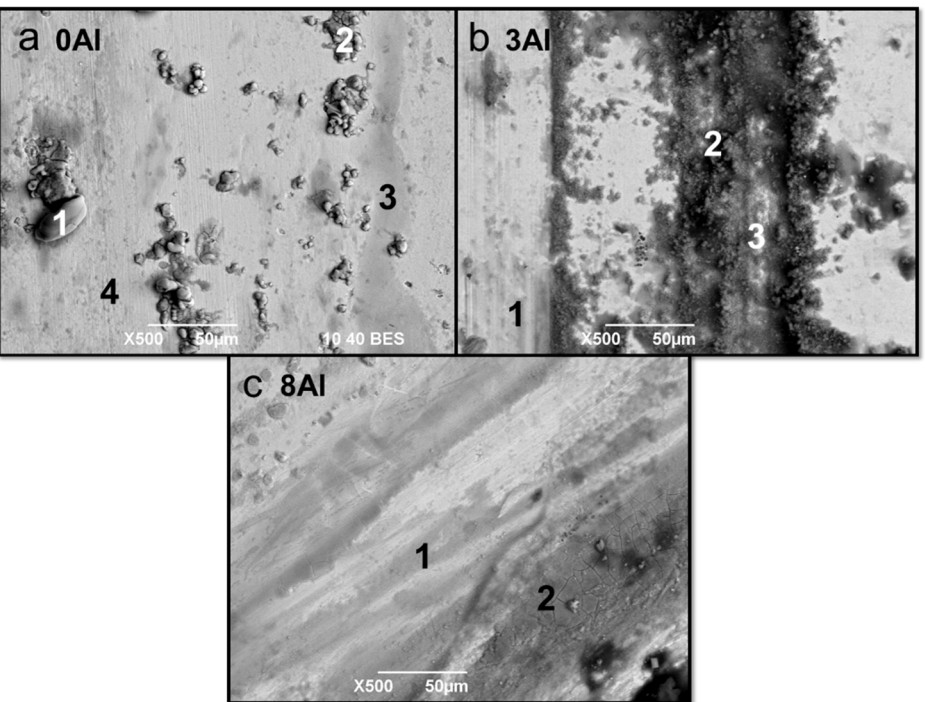

**Figure 9.** SEM images of the (**a**) 0Al; (**b**) 3Al; (**c**) 8Al alloys under total wear test, T. The numbers correspond to the spectrum showed in Table 7.

**Table 7.** EDS composition (wt.%) corresponding to the surfaces of the alloys shown in Figure 9.

| Alloy | Spectrum | C | O | Al | Cl | Ca | Na | Mn | Fe |
|-------|----------|------|-------|------|-------|------|------|-------|-------|
| 0Al | 1 | - | 29.67 | - | 10.98 | - | - | 14.32 | 45.03 |
| | 2 | 5.69 | 24.64 | - | - | - | - | 14.54 | 55.13 |
| | 3 | 7.70 | 14.28 | - | - | - | 2.05 | 15.65 | 60.32 |
| | 4 | 3.95 | 7.84 | - | - | - | 1.05 | 17.80 | 69.36 |
| 3Al | 1 | 6.62 | 15.11 | 2.28 | - | - | 2.73 | 15.77 | 57.47 |
| | 2 | 48.59 | 17.58 | 0.52 | 0.71 | 7.99 | 1.74 | 5.49 | 17.36 |
| | 3 | 14.63 | 11.62 | 1.62 | - | 3.88 | 1.20 | 17.25 | 49.80 |
| 8Al | 1 | 29.60 | 19.91 | 2.86 | 2.81 | - | - | 8.83 | 35.15 |
| | 2 | 15.32 | 24.73 | 4.24 | 1.44 | - | 4.56 | 9.53 | 40.18 |

Total wear rates are shown in Figure 10. The contribution of pure sliding wear, corrosion, and the synergistic effect between both phenomena to total wear are also shown.

Wear rates increased when compared with the pure sliding wear test indicating that corrosion reduces the wear resistance of the materials in the evaluated system. The 3Al alloy presented the highest wear rate and the 0Al alloy showed the lowest value. For all cases, the greatest contribution to the total wear rate was due to pure sliding wear, which indicates that pure mechanical wear is the predominant factor.

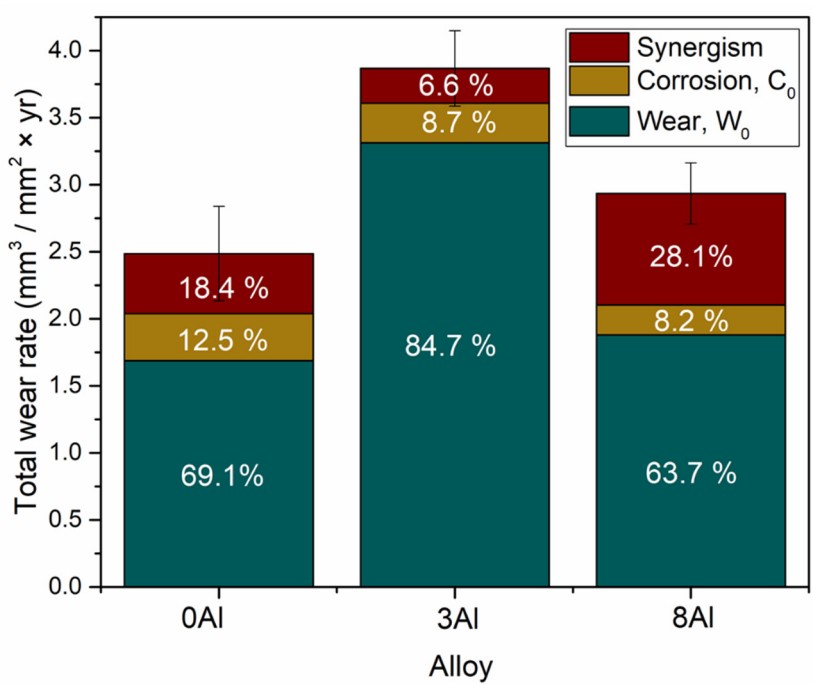

**Figure 10.** Total wear rates and contribution of pure sliding wear, corrosion, and the synergistic effect.

Figure 11 exhibits the synergy behavior of the alloys. It is observed that the 8Al alloy showed a greater synergistic effect, therefore, this alloy is more sensitive to the synergy effect than the other alloys. This effect might be due to the destabilization of the aluminum oxide layer and the interaction with the deformation mechanisms that predominate in this material, which are the formation of twins and microbands. These microbands intersect as the plastic deformation increases and subdivide the grain, which results in a grain refinement effect [47], increasing the density of grain boundaries where corrosion has a greater tendency to occur since they have higher associated energy due to the accumulation of dislocations by deformation [48,49] This further destabilizes the passive layer, which has been removed by friction, retarding its reformation process, and causing further dissolution of Fe and Mn atoms. This hypothesis is tested below with the corrosion experiments under the effect of wear.

*3.6. Corrosion + Wear Measurements, $C_w$*

Figure 12 shows the polarization curves and Table 8 shows the values of the corrosion potentials and corrosion rates with the action of wear, capital letters indicate the group according to ANOVA analysis. Regarding the OCP, there was no significant variation in the 0Al alloy, with and without wear, which contrasts with the evolution of the OCP during the total wear test shown in Figure 8a. The 3Al and 8Al alloys showed a potential shift towards more negative values due to the action of wear and the 8Al alloy showed the greatest difference in corrosion potential, which suggests that a greater galvanic coupling can occur between the worn and unworn areas increasing the trend to corrosion in the anodic areas (wear track). Corrosion rates did not change significantly with the action of wear for the three alloys.

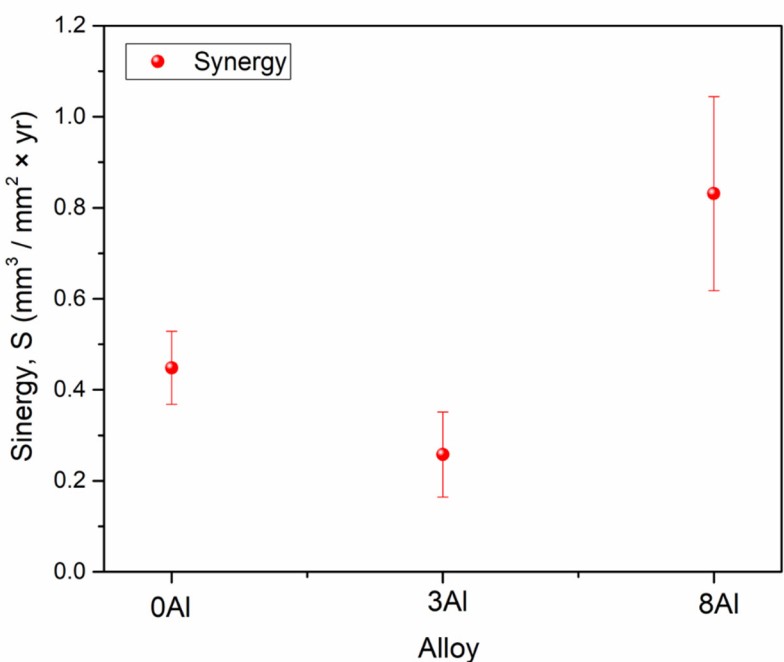

**Figure 11.** Synergy behavior of the alloys.

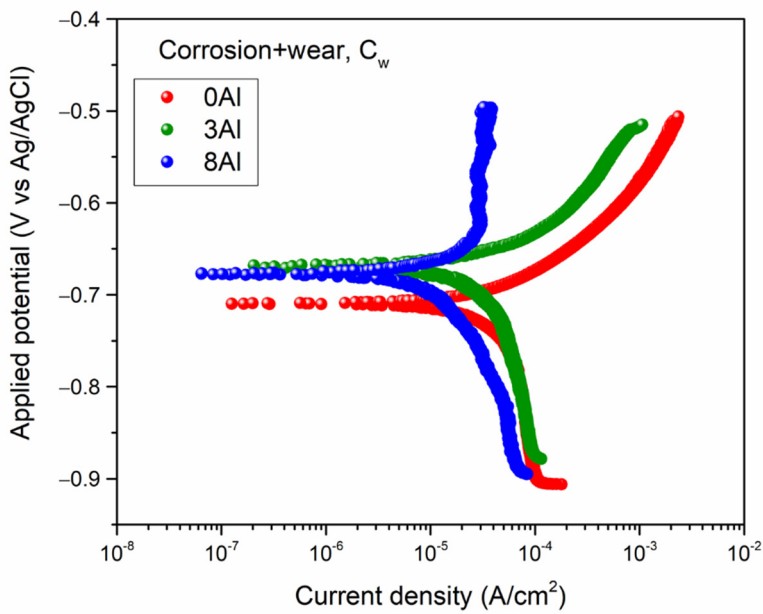

**Figure 12.** Polarization curves for the alloys corresponding to $C_w$ tests.

**Table 8.** Corrosion potentials values (compared to the Ag/AgCl reference electrode) and corrosion rates of the materials.

| Alloy | Corrosion Potential $E_{corr}$ (V vs. Ag/AgCl) | Group [1] | Corrosion Rate (mm³/mm² yr) | Group [1] |
|---|---|---|---|---|
| 0Al | $-0.71 \pm 0.01$ | A | $0.40 \pm 0.04$ | A |
| 3Al | $-0.66 \pm 0.07$ | B | $0.40 \pm 0.10$ | A, B |
| 8Al | $-0.67 \pm 0.01$ | C | $0.24 \pm 0.10$ | B |

[1] Groups according to ANOVA analysis.

According to the literature [14,50], the increase in the SFE suppresses the formation of deformation twins (i.e., twining formation must be lower in the 8Al alloy). As twins form

due to deformation, they act as an obstacle for dislocations movement, producing a local inhomogeneity due to higher density of dislocations, thereby increasing the energy and giving rise to a greater tendency to corrosion. In this sense, in the 0Al alloy (lower SFE), more twining would form during the deformation and therefore its behavior against wear and corrosion is lower than the other two alloys as shown in Table 8.

The result found in the Table 8 for the 8Al is not consistent with the hypothesis mentioned above since it would be expected that this alloy would present an increase in the corrosion rate due to its greater synergistic effect. Thus, in order to gain a better understanding of the observed behavior, it is necessary to analyze the contributions to the synergy of the change in wear rate due to corrosion, $\Delta W_c$, and of the change in corrosion rate due to mechanical wear, $\Delta C_w$. The results are shown in Figure 13. The main contribution to the synergy in all cases was due to the change in the wear rate due to corrosion and was greater for the 8Al alloy. This result may occur since the corrosion products generated and those which adhered to the counterbody tend to fracture and detach from it, becoming trapped between the surface of the material and the counterbody and then generating abrasion.

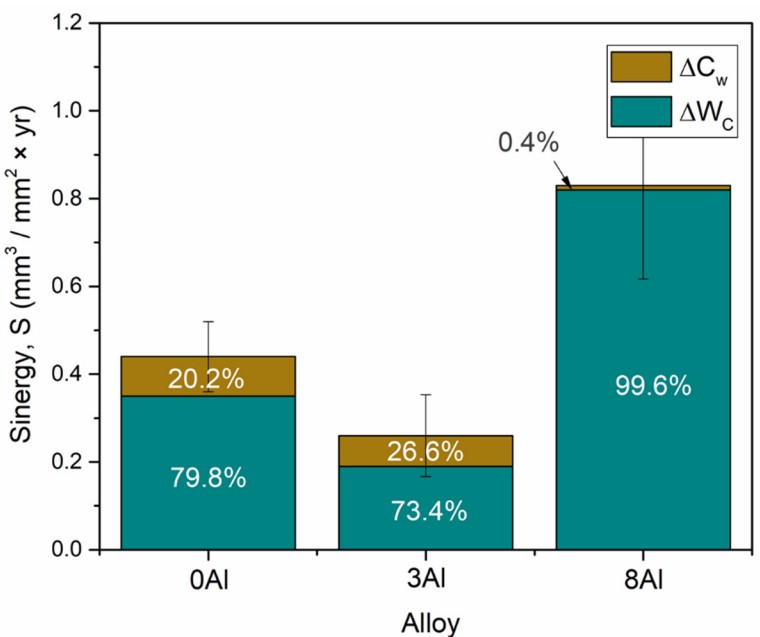

**Figure 13.** Contributions to the synergy of the change in wear rate due to corrosion, $\Delta W_c$, and of the change in corrosion rate due to mechanical wear, $\Delta C_w$.

Concerning the micro-hardness, Figure 14 shows the hardening profiles after pure and total wear tests. A remarkable increase in micro-hardness was observed after the tests compared to the initial micro-hardness in both cases of wear for the three alloys, which indicates the formation of a hardened layer due to plastic deformation. It was also seen that the depth of the hardened layer for the 3Al alloy was greater in the total wear test (about 94 μm) than for the pure wear test (about 55 μm), while in the other two alloys no significant variation was detected, indicating that corrosion had an effect on the removal of the hardened layer in the 3Al alloy.

This behavior is produced by the lower and non-uniformity of the protective and stable oxide film as well as other type of oxides formation during total sliding wear test in the 3Al alloy, causing the exposure of the base material in some areas to the electrolyte during each pass of the counterbody. Then, the elimination of metallic particles from the deformed layer mainly occurred in areas where there could be surface and subsurface cracks, generating debris that become trapped at the slip interface. This process is repeated during the continuous friction of the counterbody over the same area, giving rise to a new contact with the base material at a greater depth and producing the hardening of the base

material at greater depths. Moreover, Mao et al. [51] analyzed the grain morphology under the wear track of an austenitic alloy and identified shear band intersections in the grains. They explain that this can act as the nucleation site for stress-induced martensite [52], which would explain the change in hardness observed in Figure 14.

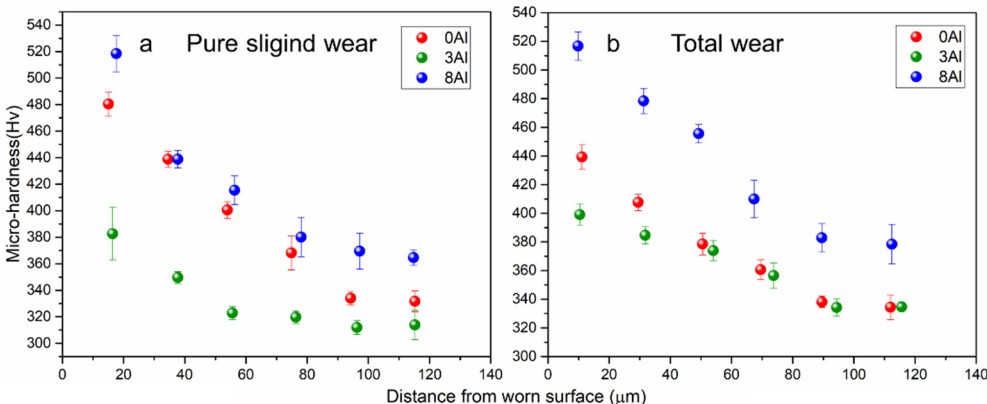

**Figure 14.** Hardening profiles of the alloys after (**a**) pure sliding wear and (**b**) total wear tests.

An investigation of Zambrano et al. [8] of the effect of normal load on abrasive wear behavior of Fe-Mn-Al-C and other austenitic steels found that the initial micro-hardness was not a useful parameter to predict the abrasive wear resistance on these materials. The results of the wear rate obtained in the present work agree with the previous finding, e.g., the initial micro-hardness of the present alloys is not useful to predict neither the pure sliding wear nor the total sliding wear resistance. Instead, it is convenient to compare the micro-hardness of the worn surface and the base material ($H_{surface}/H_{core}$). The ratio $H_{surface}/H_{core}$ for each material as a function of the pure and total wear rates is shown in Figure 15.

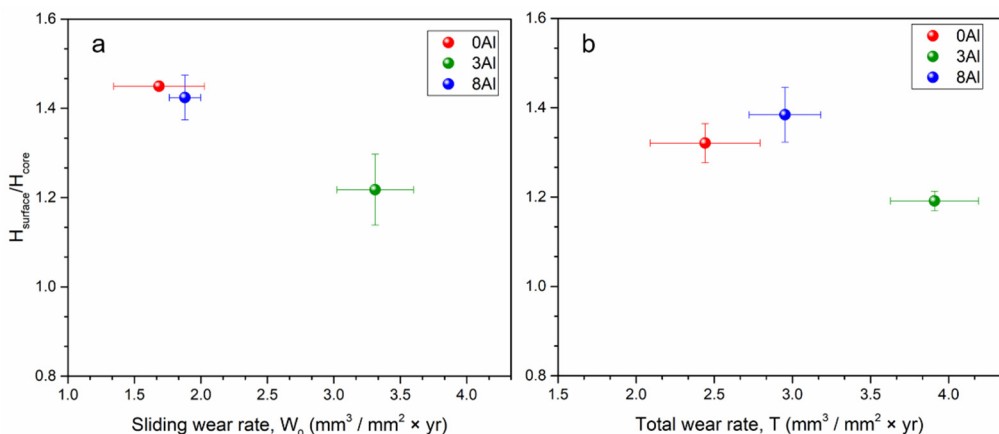

**Figure 15.** $H_{surface}/H_{core}$ value for each material as a function of (**a**) pure sliding wear and (**b**) total wear rates.

In both cases, the value of $H_{surface}/H_{core}$ was greater than one, which indicates an increase in micro-hardness due to the plastic deformation process. It was also seen that for the 0Al and 8Al alloys the ratios were not significantly different, while the 3Al showed a significantly lower value, which is related to the highest pure and total wear rates. This behavior occurs because the hardened surface layers resulting from the plastic deformation process cannot be supported by the softer base material, producing a greater removal due to the consecutive movement of the counterbody over the base material and hard oxides adhered to this [53].

### 4. Conclusions

This research analyzed the tribocorrosion behavior of Fe-Mn-Al-C alloys and its relationship with the Al content. From the results found above the following conclusions can be highlighted:

1.  The 0Al alloy showed the highest tendency to corrosion and the 8Al alloy showed the lowest tendency to corrosion and corrosion rate, which is due to the fact that Al reduces the ability to form other type of oxides and impurities and promotes the formation of a stable oxide on the surface.
2.  Total wear rates increased compared to the pure sliding wear test, indicating that corrosion reduces the wear resistance of the materials in the evaluated system. In addition, for all materials, the largest contribution to the total wear rate was due to pure sliding wear.
3.  The change in the wear rate due to corrosion provided the main contribution to the synergism and was greater for the 8Al alloy.
4.  An increase in micro-hardness after wear (total and pure) with respect to the initial micro-hardness was observed, which indicates the formation of a hardened layer produced by plastic deformation. The layer was not supported by the base material, mainly in the 3Al alloy, producing a greater removal of the material due to the consecutive friction with the counterbody.

**Author Contributions:** G.M.B.-O.: conceptualization, methodology, investigation, writing—original draft preparation; L.A.T.: conceptualization, writing—review and editing, supervision; S.A.R.: project administration, writing—review and editing, supervision; J.J.C.: writing—review and editing, supervision. All authors have read and agreed to the published version of the manuscript.

**Funding:** This research was founded by MINCIENCIAS through project No. 1106-808-63096.

**Institutional Review Board Statement:** Not applicable.

**Informed Consent Statement:** Not applicable.

**Data Availability Statement:** Not applicable.

**Acknowledgments:** The authors acknowledge to MINCIENCIAS and Universidad del Valle for the support obtained through project No. 1106-808-63096.

**Conflicts of Interest:** The authors declare no conflict of interest.

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
