# Peer review of "On the Tribocorrosion Behavior of Fe-Mn-Al-C Alloys in Ringer’s Solution"

_metals, doi:10.3390/met12081339_

Round 1

Reviewer 1 Report

Recommendations

Manuscript ID: metals-1822695.

Type of manuscript: Article.

Title: On the tribocorrosion behavior of Fe-Mn-Al-C alloys in Ringer’s solution.

Abstract: Very well written and conveys the main points that represent the article.

Introduction: The bibliographic research is very well supported.

2. Materials and Methods  

As a complement, it would be advisable to give the name of the piece of equipment employed to perform the petrochemical analysis.

2.1 Tribocorrosion test

Reference 27 is wrongly cited. It should be checked.

The authors should describe or indicate how the reported surface roughness was measured (Ra=0.03 ± 0.07 μm). It is not clear how it was obtained.

Figures 1 (a) and (b) display the same schematic information. Just one figure should be considered.

It is clear that Icorr and Ecorr were obtained from the Tafel curves, but how were these values computed? This question is prompted by the fact that these curves did not display Tafelian behavior in order to obtain the Tafel slopes. This point should be included in the manuscript. 

3. Results and Discussion

3.1. Microstructural characterization 

It is necessary that the authors indicate under what norm the grain size was measured. In this sense, the following is stated: “...grain size is not a relevant factor in the discussion…” This statement should be analyzed, otherwise, the set research goal, whose analysis is clear, is not reached and Section 3.1 would be without any contribution to the work.

In another part, it is written: “Some casting defects...Some casting defects (black spots) ... which were identified by using EDS (Energy Dispersive X-ray Spectroscopy) as microporosities and iron or manganese oxides, as well as pitting caused by chemical attack.” Notwithstanding, there is no evidence to support such statement, for the corresponding EDS spectra are nowhere to be seen.

Also, the following was speculated: “On the other hand, there are two hypotheses for the increase in micro-hardness in...” To my mind, in all scientific work, once hypotheses have been set, the next step is to work to get scientific evidence either to confirm or discard them, but what cannot be done is to repeat them. This fact happened in another manuscript section. It should be kept in mind that in order to have scientific contributions, hypotheses should be set carefully to be confirmed with scientific evidence.

3.3 Corrosion measurements, C0

The authors state that Ecorr displays displacements toward less negative potentials for the analyzed materials, however, there is no deep explanation in sight as to what prompted such a phenomenon. What interphase phenomena took place? What role do the material active sites play? What caused the behavior of the current density in the Tafel branches? How were Icorr and corrosion rate calculated if the curves did not display Tafelian behavior?

3.4. Pure sliding wear measurements, W0

The authors wrote: “The particles deposited... are result of the cathodic protection…which produces a reduction in water and the production of OH- ions.” I wish these products (chemical formulas) capable of favoring such cathodic protection properties were mentioned in the manuscript. Furthermore, are these the only ions in the aqueous medium? In the case of having a protecting layer on the surface, in what part of the Tafel plots can this fact be confirmed? The answers to these questions would support the authors’ discussion employing the electrochemical technique.

Reviewer 2 Report

Interesting work that is presented in a logical way. Found few grammatical errors and recommend a proof read. 

Reviewer 3 Report

The tribocorrosion behavior of Fe-Mn-Al-C alloy was systematically studied. It can be considered to publish after solving the following problems.

1. The research background of wear and corrosion behavior of Fe-Mn-Al-C alloy needs to be further strengthened in INTRODUCTION.

2. The author mentioned the effect of Al content on SFE, but it seems that no further research has been carried out. Therefore, the relationship between SFE and corrosion, wear and synergistic effect needs to be further explained.

3. The data points of three alloys with different Al content are relatively few. Can the author add two more data points of Al content to fully explain the influence trend of Al content on corrosion and wear behavior.

4. It is suggested that the author provide the longitudinal section grain morphology after the test and explain the hardness change in combination with figure 13.

5. Some unnecessary conclusions need to be deleted as far as possible.
